# The Impact of Nickel–Zinc Ferrite Nanoparticles on the Mechanical and Barrier Properties of Green-Synthesized Chitosan Films Produced Using Natural Juices

**DOI:** 10.3390/polym16243455

**Published:** 2024-12-10

**Authors:** Dilawar Hassan, Ayesha Sani, Aurora Antonio Pérez, Muhammad Ehsan, Josué D. Hernández-Varela, José J. Chanona-Pérez, Ana Laura Torres Huerta

**Affiliations:** 1Tecnologico de Monterrey, School of Engineering and Sciences, Atizapan de Zaragoza C.P. 52926, Estado de Mexico, Mexico; a01754343@tec.mx (D.H.); a01754344@tec.mx (A.S.); a.antonio@tec.mx (A.A.P.); 2Centro de Bachillerato Tecnológico Agropecuario 162. Carr. Mexico-Veracruz vía Texcoco km 95, Francisco I. Madero C.P. 90280, Tlax, Mexico; muhammadehsan2000@yahoo.com; 3Laboratorio de Micro y Nanobiotecnología, Departamento de Ingeniería Bioquímica, Escuela Nacional de Ciencias Biológicas, Instituto Politécnico Nacional, Av. Wilfrido Massieu s/n, Mexico City 07738, Mexico; jhernandezv1717@alumno.ipn.mx (J.D.H.-V.); jchanona@ipn.mx (J.J.C.-P.)

**Keywords:** chitosan, green polymers, green chemistry, natural, polymeric film

## Abstract

A trend has been established concerning the research and development of various green and biodegradable plastics for multi-purpose applications, aiming to replace petroleum-based plastics. Herein, we report the synthesis of chitosan (CH) films using lemon juice; these were reinforced with NiZnFe_2_O_4_ nanoparticles (NiZnFe_2_O_4_ NPs) to obtain improved mechanical and barrier properties, facilitating their future application as sustainable, corrosion-resistant coatings for medical instruments. The synthesized NiZnFe_2_O_4_ NPs had a crystallite size of ~29 nm. Reinforcement with the nanoparticles in bio-sourced chitosan films was conducted at two concentrations: 1% and 2%. The mechanical strength of the CH film was found to be 1.52 MPa, while the 2% NiZnFe_2_O_4_ NP-containing films showed stress-bearing potential of 1.04 MPa with a larger strain value, confirming the elastic nature of the films. Furthermore, the % elongation was directly proportional to the NP concentration, with the highest value of 36.833% obtained for the 2% NP-containing films. The CH films presented improved barrier properties with the introduction of the NiZnFe_2_O_4_ NPs, making them promising candidates for coatings in medical instruments; this could protect such instruments from corrosion under controlled conditions. This approach not only broadens the application range of biopolymeric films but also aligns with global sustainability goals, serving to reduce the reliance on non-renewable corrosion-resistant coatings.

## 1. Introduction

Plastics are widely utilized across various industries due to their transparent and flexible nature, making them ideal for applications such as surface coatings to protect against corrosion [1,2]. They are predominantly sourced from petrochemical-based non-renewable materials [3], such as polypropylene (PP), polyethylene (PE), and polyethylene terephthalate (PET), which constitute over 80% of all plastic production, particularly for packaging [4] and surface protection [2]. This extensive use has led to significant environmental stress due to their slow degradation rates [5]. Additionally, the transformation of these plastics into micro- and nano-plastics has led to a new area in toxicity research. Microplastics have been linked to various diseases and have been detected in the human lungs [6] and in fish gills and guts [7]. After reaching the micro scale, these plastics are slowly converted into nano-plastics, becoming harmful to humans [8], marine life [9], and ecosystems [10].

To address these issues, green and sustainable methods are being explored to develop suitable alternatives to traditional plastics. Among the broad range of polysaccharides, chitosan is the most utilized biopolymer in the generation of biodegradable plastic films; it is also known to be the second most naturally abundant polysaccharide [11]. Chitosan results from the deacetylation of another compound called chitin [12]. Alongside its biodegradable nature, it is also classified by the US FDA as ‘Generally Regarded as Safe’ [13].

Traditional methods for the development of CH films utilize analytical-grade acids, which render these production methods hazardous and non-ecofriendly. As chitosan is not water-soluble, acids such as citric or acetic acid are generally used to obtain a 1% solution [14,15]. It is well known that the production of glacial citric acid is achieved with the use of sulfuric acid [16]. To address these issues, we utilize lemon juice as a replacement and a sustainable alternative to analytical-grade acids, ensuring the utilization of the entire lemon.

To enhance the barrier properties and mechanical properties of the films and to achieve better corrosion resistance, a variety of reinforcements have been incorporated into the matrix [17,18,19]. Meanwhile, some studies have incorporated nanoparticles into the polymeric film to investigate their effects on its flexibility [20]. For instance, J. Ashfaq and his group developed a PVA film reinforced with graphene, and the composite film showed mechanical strength of 341 MPa, as compared to a blank CH film’s strength of 165 MPa [21]. Another group, led by J. A. A. Abdullah, reported the development of CH films reinforced with iron oxide nanoparticles. This group reported that the iron oxide nanoparticles acted as nanofillers and caused the CH structure to stiffen, resulting in enhanced mechanical properties and mechanical strength of 9.2 MPa, as compared to the neat CH film’s strength of 8.2 MPa. Therefore, the modification of the films’ properties depends on the dispersion of the nanoparticles in the film structure. Homogeneous dispersion increases the flexibility and strength by reinforcing the film via the nanoparticles [22], while non-homogeneous dispersion damages the polymeric structure, forming voids that lower its load-bearing potential. Various metal oxide nanoparticles are known for their biomedical applications; for instance, iron oxide [23,24], nickel oxide [25,26], and zinc oxide [27,28] nanoparticles are all FDA-approved [24,29]. Their composites, in the form of NiZnFe_2_O_4_ NPs, are also known for their magnetic properties and biomedical applications [30]. They have been used for their potential supercapacitance [31], in electronic applications [32], in drug loading for cancer therapy [33], and in antimicrobial applications [34]. Furthermore, reinforcements with NiZnFe_2_O_4_ NPs extend the application of such films to the fields of surface engineering and tribology. Improving their mechanical strength and elasticity enables these films to be used as sustainable bio-based coatings for the protection of surgical and other mechanical components from wear, corrosion, and other types of surface deterioration.

Extracts from plants and vegetables have been used by many researchers in an effort to further enhance the antioxidant, antibacterial, and antifungal properties of chitosan films [35]. One fruit that exhibits health benefits is lemon. Lemon is known to contain many phytochemicals, such as terpenoids, alkaloids, and flavonoids [36]. The pectin in its peel has been utilized in several studies to prepare chitosan-based polymeric films [37], with applications ranging from food to storage. These bio-compounds further add to the barrier properties of the films; for instance, H. Aloui et al. developed sodium alginate films reinforced with gallnut extract and reported that these films showed almost 1.7 times lower water vapor permeability per day compared to pristine sodium alginate films. However, due to the hydrophobic nature of bio-compounds, they caused a decline in mechanical strength. Nonetheless, in turn, they enhanced the flexibility of the films, as reported by P. Rachtanapun’s group, where the observed elongation-at-break value was 49.45% after adding a curcumin extract, as compared to 17.74% for the neat CH film [38].

In this work, we avoided using analytical-grade glacial acids for chitosan dissolution and instead dissolved it in lemon juice, which contains approximately 1.44 g/oz citric acid [39], seeking to develop a greener approach. In addition, we utilized lemon peel extract for the production of NiZnFe_2_O_4_ nanoparticles, aiming to ensure the use of the entire fruit. The greener approach makes this a novel method, as there are no studies that have used lemon juice as a replacement for citric acid in the dissolution of CH. Furthermore, this method is sustainable, as lemon peel is used to obtain LPE for the synthesis of reinforcing materials, i.e., NiZnFe_2_O_4_ nanoparticles. The chitosan films were reinforced with NiZnFe_2_O_4_ nanoparticles to enhance their mechanical and barrier properties. In this investigation, we evaluated various attributes, including the mechanical strength, puncture resistance, elongation at break, water vapor permeability, water solubility, moisture content, and degree of swelling, comparing both nanoparticle-reinforced and bare chitosan films. The results could facilitate the fulfillment of both industrial needs and sustainability goals.

## 2. Materials and Methods

Lemons (*Citrus aurantifolia*) were purchased from a local market close to Tecnologico de Monterrey, Villas de la Hacienda, Estado de Mexico. Precured salts (zinc acetate—Zn (CH_3_CO_2_)_2_, iron sulfate heptahydrate—FeSO_4_·7H_2_O, and nickel acetate tetrahydrate—Ni(COOCH_3_)_2_·4H_2_O); chitosan 50–190 kDa; glycerol; and 1,1-diphenyl-2-picrylhydrazyl were purchased from Merck, Toluca, Mexico, and were all of analytical grade.

### 2.1. Lemon Juice and Lemon Peel Extraction

The lemons were visually inspected for deformities or signs of softness, with any compromised fruits excluded from the study. The chosen lemons were thoroughly washed with tap water, followed by a rinse with distilled water to remove surface contaminants. After air drying at room temperature, the lemons were halved. Lemon juice was extracted using a pre-washed and autoclaved handheld lemon juice squeezer and collected into a beaker. The juice was then filtered through Whatman #40 filter paper to remove suspended fibers or particulates. The clarified juice was stored at 4 °C in airtight storage bottles for further use over the next two weeks. After extracting the lemon juice, the residual lemon peels were gathered and drenched in a solution of 10 g lemon peel in 100 mL distilled water. The extraction of the lemon peel extract (LPE) was performed based on a modified process that has been previously described [24]. On a hot plate, the temperature was set at 60 °C with constant stirring to allow extraction over a period of four hours. Afterward, the solution was filtered three times using Whatman #40 filter paper (Merck, Toluca, Mexico). The filtered extract was kept at 4 °C in airtight storage bottles for subsequent use.

### 2.2. Nanoparticle Synthesis

A slightly modified version of a previously established method was used for the synthesis of NiZnFe_2_O_4_ [26,40]. In brief, a mixture of nickel acetate (0.5 g), zinc acetate (0.5 g), and iron sulfate (1 g) salts at a ratio of 0.25:0.25:0.5 was prepared in a 50 mL beaker. Then, it was heated on a hot plate under constant stirring while being maintained at 60 °C. At the same time, 100 mL of LPE was heated to 60 °C on another hot plate under continuous stirring. One hour later, the salt mixture was added to the LPE-containing beaker. The temperature of the combined solution was raised to 80 °C. The reaction was maintained for 4 h, with constant stirring at 800 rpm. After completion, it was centrifuged (using an Eppendorf 5430R centrifuge, Hamburg, Germany) at 10,000 rpm for 15 min. The precipitates obtained were washed with distilled water and dried in an oven at 100 °C for 3 h. Afterward, the precipitates obtained were dried and annealed inside a muffle furnace at 450 °C for three hours. The resulting NiZnFe_2_O_4_ NP powder was kept in hermetically closed falcon tubes for subsequent analysis and application. Figure 1a shows a schematic of these processes, from lemon peel extraction to nanoparticle synthesis.

### 2.3. Nanoparticle Characterization

The synthesized nanoparticles, designated as NiZnFe_2_O_4_ NPs, were characterized using various techniques. The crystalline phase of the biosynthesized nanoparticles was identified using a Bruker D-8 advanced X-ray diffractometer (Karlsruhe, Germany). XRD is a basic technique used to assess crystalline structures. The D-8 device was equipped with Kα of Cu, with λ = 1.5 Å. The Scherrer formula was employed to determine the crystallite sizes of the biosynthesized NPs [41]. The surface morphologies of the biosynthesized NiZnFe_2_O_4_ nanoparticles were examined using a JEOL JSMIT700 HR scanning electron microscope (Tokyo, Japan). The accelerating current was set to 10.0 kV, and the working distance (WD) was set at 41.3 mm. SEM images are crucial in understanding the dispersion and aggregation behaviors of nanoparticles, as they explicitly indicate whether they are clustered or exist in dispersed individual forms, possibly due to their non-magnetic nature. The functional groups attached to the nanoparticles derived from LPE were analyzed in the range of 4000–400 cm^−1^ using a Spectrum-2 Fourier-transform infrared spectrometer (FTIR) by PerkinElmer (Shelton, CT, USA). Given that the nanoparticles were synthesized using LPE, it was expected that they would incorporate biomolecules from the extracts. These biomolecules not only stabilized the nanoparticles but also inhibited further nucleation.
*D_p_* = *K*λ/*β*cos*θ*(1)
where

*D_p_* = the crystallite size of the nanoparticles;

*K* = the Scherrer constant;

λ = the wavelength of incident waves;

*β* = the full width at half maximum (FWHM).

### 2.4. Chitosan Film Development

The films were fabricated using ratios of 0.2:0.8, 0.25:0.75, and 0.3:0.7 (*v*/*v*) of lemon juice and LPE, respectively. A 2% (*w*/*v*) concentration of CH was used. The mixture with the 0.2:0.8 ratio led to the precipitation of CH at the bottom of the beaker, which was maintained at 40 °C under constant stirring. Although the 0.25:0.75 ratio led to the near dissolution of all the CH, the 0.3:0.7 ratio was ultimately selected based on optimization experiments that indicated the complete dissolution of CH under these conditions. In the preparation process, 2 g of CH was introduced into a 100 mL mixture of lemon juice and LPE (30 mL and 70 mL, respectively), which was pre-heated. The mixture was stirred to facilitate CH dissolution over a period of 30 min. Subsequently, glycerol was added at a concentration of 1.5% relative to the weight of CH, and the mixture was stirred at 900 rpm for an additional 20 min at 40 °C. To eliminate any entrapped air bubbles, the stirring speed was reduced to 400 rpm for 20 min. Finally, 40 mL of the homogeneous solution was evenly distributed into each Petri dish and dried in a hot air oven at 45 °C for 30 h to allow the formation of plastic films. A general schematic is shown in Figure 1b.

### 2.5. Development of Chitosan Films Containing Nanoparticles

For the reinforcement of the NiZnFe_2_O_4_ NPs, the optimized concentrations of lemon juice and LPE were used with 2 g of CH, maintained under constant stirring at 40 °C. The concentrations of 1% and 2% NiZnFe_2_O_4_ NPs (*w*/*w* percent of CH) were tested before adding glycerol. After mixing the NiZnFe_2_O_4_ NPs, the stirring was continued for another 20 min. Glycerol was then added to the mixture, and it was stirred for another 20 min. Finally, 40 mL of the CH and NiZnFe_2_O_4_ NP solution was poured into Petri dishes and dried in a hot air oven at 45 °C for 30 h.

### 2.6. Physiochemical Characterization of Chitosan Films

Several chitosan films, with and without ZnFe_2_O_4_ nanoparticles, were characterized to evaluate their properties. The analyses included scanning electron microscopy (SEM), Fourier-transform infrared spectrometry (FTIR), and atomic force microscopy (AFM). A texture analysis was performed to assess mechanical parameters such as the tensile strength, puncture resistance, and elongation at break. The films were further checked for their moisture content, degree of swelling, solubility in water, water vapor permeability, and optical characteristics, including the color of transmittance.

The JEOL model JSM-IT700 HR was utilized to conduct a topographical analysis of the developed films. The purpose of this was to analyze the changes in the surfaces of the films both with and without the introduction of NiZnFe_2_O_4_ NPs. Images were taken using an acceleration current of 3.0 kV at a working distance (WD) of ~42 mm. Additionally, for the 3D surface analysis, the Park Systems XE7 (Park System, Suwon, South Korea) atomic force microscope was employed. For this analysis, a 1 × 1 cm square piece of film was cut and used without any modifications. The PerkinElmer Spectrum-2 Fourier-transform infrared spectrometer was utilized to analyze the functional groups present in the CH films and those containing nanoparticles. A small piece of each film was placed in the FTIR instrument and analyzed across the spectral range of 4000–500 cm^−1^. The resulting spectral data were plotted using Origin Pro software 2021 (64-bit, version 9.8.0.200). The main reason for this evaluation was to detect the presence of functional groups from lemon juice and LPE, whereas FTIR was employed to determine the presence of a new peak when the nanoparticles were incorporated inside the polymeric matrix.

The thickness of each film (n = 3) was measured using a vernier caliper. Each film sample was placed between the jaws of the caliper, and a reading was recorded. Subsequently, readings were taken from two additional, randomly selected spots on the same film. In total, three readings were obtained for each film sample, and the average of these three readings was calculated to determine the average thickness. The film thickness is a major factor that can play a role in the mechanical properties [42] and physical appearance [43] of films. Color analysis is also a major tool in the study of films. Lighter colors are mostly used to appeal to customers, as the product can be seen through them. For the color analysis of the films, images of all films were taken from a constant distance under standardized lighting conditions with a Canon camera. ImageJ 1.53k software was then used to process these images (NIH—USA) [44]. Each image was cropped to 300 × 300 pixels by selecting the center of the film and processed to obtain the *L**, *a**, and *b** values. *L** defines the lightness and darkness of the film, where a positive value indicates lightness and vice versa; *a** defines the shift from red to green, with positive values showing a red shift; and *b** defines the shift from yellow to blue, with positive values representing yellow. Δ*E* quantifies the color difference between the displayed and reference images (a white background where the photo was captured), with values ranging from 0 (no difference) to 100 (complete distortion); it is defined by the following equation [44]:(2)$$∆E={\left[{\left({∆L}^{*}\right)}^{2}\right]+\left[{\left({∆a}^{*}\right)}^{2}\right]+\left[{\left({∆b}^{*}\right)}^{2}\right]}^{1/2}$$
where

∆*L** = *L** − *L**_0_ (*L**_0_ = reference value and *L** = value for film sample);

∆*a** = *a** − *a**_0_ (*a**_0_ = reference value and *a** = value for film sample);

∆*b** = *b** − *b**_0_ (*b**_0_ = reference value and *b** = value for film sample).

#### 2.6.1. Moisture Content (MC), Water Solubility (WS), and Degree of Swelling (DS)

The moisture content (MC), water solubility (WS), and degree of swelling (DS) are factors that define the degradability and stability of a film. Higher MC, DW, and WS values result in softer and more degradable polymeric films, and such films cannot be recommended for applications in moist environments. For the measurement of the %MC, %WS, and %DS, square pieces of each film sample measuring 20 × 20 mm were cut and weighed; this initial weight was recorded as M_1_. The film samples were then placed in a hot air oven set at 70 °C and left for 24 h to eliminate any retained moisture. After this period of drying, the samples were again weighed to obtain the M_2_ value. Then, the dried film samples were placed in a Petri dish filled with 45 mL of distilled water for 24 h to absorb moisture. The weight was measured after removal from the Petri dish and surface drying to determine M_3_. The film samples were again placed in the hot air oven at 70 °C for another 24 h to determine the M_4_ value. The following formulas were then used to calculate the respective values [44]:(3)$$\mathrm{Moisture}\;\mathrm{content}\;(\%)=\frac{M_{1}-M_{2}}{M_{1}}\;\times \;100$$
(4)$$\mathrm{Water}\;\mathrm{solubility}\;(\%)=\frac{M_{2}-M_{4}}{M_{2}}\;\times \;100$$
(5)$$\mathrm{Swelling}\;\mathrm{degree}\;(\%)=\frac{M_{3}-M_{2}}{M_{2}}\;\times \;100$$

#### 2.6.2. Water Vapor Permeability

A water vapor permeability (WVP) study was performed to determine the applications of the prepared films. For instance, the medical and food industries require polymers that have lower WVP values, which indicate stronger barrier properties, so that the product exhibits greater safety and its quality can be maintained. A previously employed method was used [45]. Briefly, 10 mL of water was placed inside a test cup, and the film sample was placed over the cup’s surface to ensure leak-proof conditions. This was achieved using vacuum grease, which was further covered with a steel ring tightened with the help of screws to provide better sealing. The film-sealed cups were then placed in a desiccator with RH = 75%, followed by placing the desiccator in a controlled-environment chamber (Thermo Scientific (Waltham, MA, USA), 3911—Forma Environmental Chamber). The weight of the cup was recorded prior to placement in the desiccator, and the changes in weight were measured every 24 h over a period of three days [46] to determine the amount of water vapor transmitted through the film. The following equation was used to measure the WVP [45]:(6)$$WVP=(\frac{∆w\;\cdot \;\;l}{A\;\cdot \;∆P\;\cdot t})$$
where

∆*w* = the weight difference (g);

*l* = the film thickness (m);

*∆P* = the difference in vapor pressure at 25 °C (3170 Pa);

*A* = the film area (m^2^);

*t* = the permeation time (s).

The *W_f_* value computed for the first 24 h was taken as *W_i_* for the reading taken after 72 h.

Furthermore, the aspect ratios of the 1% and 2% NiZnFe_2_O_4_ NPs in the films were determined by rearranging Bhardwaj’s permeability model [47], giving the following equation:(7)$$A=\frac{L}{W}=\frac{3\;\left(1-\phi -\frac{P_{c}}{P_{m}}\;\right)}{\phi \;\cdot \;\frac{P_{c}}{P_{m}}\;\cdot \;\left(S+\frac{1}{2}\right)}\;$$
where

*P_c_* = the permeability of the composite material;

*P_m_* = the permeability of the pristine CH film;

*S* = the orientation or shape factor;

*ϕ* = the volume fraction of the nanoparticles in the matrix;

*A* = the aspect ratio of the nanoparticles.

In Bhardwaj’s permeability model, if the tortuosity factor α approximates the aspect ratio *A*, the formula is simplified, as we can consider *α* ≈ *A*.

#### 2.6.3. Mechanical Properties

Knowledge of their mechanical properties helps in determining whether the developed polymeric films are suitable for industrial use. For example, in industry, heavy equipment and parts are typically used, so a soft film with poor mechanical properties would not be suitable for industrial use. The textural properties of the chitosan films and the effect of the NiZnFe_2_O_4_ NP concentration incorporated within the films were determined using a CT3-10000 texture analyzer from Brookfield, Middleboro, MA, USA. The initial cell load was 2 N, while the final grip separation value was set at 30 mm, with a crosshead speed of 0.20 mm/s. To carry out these experiments, first, each film was cut into strips with dimensions of 10 mm × 60 mm, and then each strip was gripped between T-96 double clamps. This analysis yielded four key parameters: stress (*σ*), elongation at break (%E), Young’s modulus (E), and strain (*ε*). The calculations for the stress and strain were performed using the following equations [44]:(8)$$\varepsilon \;(\%)=\left(\frac{L}{L_{0}}\right)\times 100$$
(9)$$\sigma \;(\mathrm{MPa})=\frac{F}{A_{0}L_{0}}$$
where *L*_0_ is the initial length of the sample in mm, *F* is the force at the instant that it breaks in N, *L* is the final length of the sample at the break point in mm, and *A*_0_ is the initial cross-section or thickness of the film in mm.

#### 2.6.4. Puncture Strength

A previously employed method was followed for this evaluation [48]. A needle with a 0.5 mm cross-sectional area was used to test the puncture strength of the film samples, which gave a measure of their surface toughness. This test was employed to identify the load that the film surface could tolerate before rupture. An important parameter in texture analysis and puncture tests is the distribution of the applied force. In the case of texture analysis, the force applied is distributed homogenously across the cross-sectional area of the film, while, in the case of puncture tests, the force applied focuses on a selected area. Higher surface strength indicates a stronger material that can withstand friction better and keep the coated materials safer. For this study, the same CT3-10000 texture analyzer was used. The initial load was set at 0.05 N, and the speed of the needle was 0.5 mm/s. The film samples were held in a die with a central hole with a diameter of 5 mm, and the film was clamped between the two halves of the holder using screws.

## 3. Results and Discussion

### Characterization of NiZnFe_2_O_4_ Nanoparticles

The biosynthesized and stored NiZnFe_2_O_4_ NPs were analyzed in terms of their physicochemical properties to confirm their crystalline phases and topographical features. The XRD patterns of the synthesized NPs matched the JCPDS card #52-0278 [49], as shown in Figure 2, confirming the formation of NiZnFe_2_O_4_ with a spinel structure. No additional peaks were seen in the patterns, confirming that there were no impurities in the sample. The most prominent diffraction planes identified were (111), (220), (311), (222), (400), (422), (511), and (440). The crystallite size, calculated using the Scherrer equation, was approximately 28.787 nm, as shown in Table 1.

An FTIR analysis was conducted to examine the attachment of functional groups to the surfaces of the NiZnFe_2_O_4_ NPs from the biological extract used in their biosynthesis. The attachment of functional groups can be attributed to either the entrapment within the nanostructure or the binding of biomolecules, which stabilize the newly synthesized NPs by inhibiting further nucleation. The peaks at 421 and 552 cm^−1^ correspond to Ni–O stretching [50] and metal–O vibrations [51], respectively. Additionally, the peak at 1101 cm^−1^ aligns with C–H bending [52], while the peaks at 1629 and 2110 cm^−1^ are associated with O–H bending [53] and C≡C stretching, respectively. The final peaks at 2985 and 3280 cm^−1^ correspond to C–H stretching and O–H stretching, respectively. Figure 3a shows the FTIR results for the NiZnFe_2_O_4_ NPs.

Scanning electron microscopy (SEM) was employed to investigate the morphological characteristics of the NiZnFe_2_O_4_ NPs. The SEM images, as shown in Figure 4, revealed that the NPs predominantly displayed a clustered arrangement. This clustering is primarily attributed to the intrinsic magnetic properties of the NPs, which promote aggregation [54,55]. Furthermore, the NPs exhibited distinct self-assembling behavior, organizing themselves into porous structures [24]. These pores were not merely surface features but were integrated throughout the clusters, suggesting a complex internal structure. The images clearly differentiate the NPs and pores: the NiZnFe_2_O_4_ NPs appear as bright regions due to their higher electron densities, while the darker areas represent the pores, indicating regions of lower electron density that extend from the surface into the deeper matrix of the cluster.

## 4. Physiochemical Characterization of Chitosan Films

### 4.1. Surface Morphologies of Synthesized Films Using SEM

The analysis revealed distinctive topographic variations that correlated with the nanoparticle concentrations within the films. The pure CH film exhibited a relatively smooth surface interspersed with a few dark spots. These spots were likely indicative of pores or gas bubbles trapped beneath the surface, which are common in film-casting processes [56]. Figure 5 contains the SEM images captured for all three samples. Figure 5a is an image of the CH film, and Figure 5b,c show the 1% and 2% NiZnFe_2_O_4_ NP-containing CH films. Introducing 1% NiZnFe_2_O_4_ NPs resulted in a larger number of spots, suggesting that these were not merely topographic anomalies but the NPs themselves embedded within the polymer matrix. This incorporation led to minor roughness due to the physical presence of the NPs. This roughness could also have been due to vacancies, which caused a decrease in the mechanical strength of the films. The surface of the film with 2% NiZnFe_2_O_4_ NPs was considerably rougher. This increased surface roughness could be attributed to the higher concentration of NPs, which, due to their magnetic nature, tend to clump together. This clumping results in the bunching of the NPs within the polymeric matrix, disrupting the uniformity of the surface and hence giving a more textured appearance, as shown in the figures. Furthermore, there were declines in the film surface, which could indicate collections of vacancies.

### 4.2. Surface Morphologies of Synthesized Films Using AFM

AFM was also used to confirm the surface textures of the CH films and yielded similar observations to those obtained using SEM. Figure 6a presents the smooth surface of the pure CH film, which is representative of a standard polymer film with no additives. Its smoothness may be interpreted to indicate that it had a uniform polymer matrix without remarkable interruptions or inclusions, which are characteristic of films cast from homogeneous solutions. In contrast, the films containing 1% NiZnFe_2_O_4_ NPs, as shown in Figure 6b, exhibited increased surface roughness [57]. This textural change can be attributed to the presence of NPs dispersed within the polymeric matrix. The NPs, by their nature and due to their interactions with the polymer chains, created micro-level topographic variations that were detectable with AFM. The 2% NiZnFe_2_O_4_ NP-containing films showed even greater roughness compared to the 1% NP films, as can be seen in Figure 6c. This increase in roughness was likely due to the higher density of the NPs within the film. The clustering effect, driven by the magnetic interactions among the NPs, resulted in more pronounced surface irregularities. Such roughness is not merely a surface characteristic but reflects the altered internal structure of the film due to NP aggregation.

### 4.3. Fourier-Transform Infrared Spectrometry

The Fourier-transform infrared spectroscopy (FTIR) analysis provided insightful data regarding the interactions and compatibility of the components within the CH films, especially those containing the NiZnFe_2_O_4_ NPs. The FTIR results showed notable similarities across the different film samples, highlighting specific transmittance peaks that indicated the presence and interactions of key elements. Specifically, the films containing NiZnFe_2_O_4_ NPs exhibited prominent transmittance peaks at 421 cm^−1^ and 552 cm^−1^. These peaks were characteristic of the vibrational modes associated with the metal–oxygen bonds in the NPs—specifically those involving nickel, zinc, and iron. Hence, these results are consistent with the FTIR pattern recorded for pure NiZnFe_2_O_4_ NPs [58]. This confirms that the latter were successfully incorporated into the chitosan matrix, without alterations to their intrinsic chemical structure. Furthermore, the peaks in the range of 1000 to 600 cm^−1^ represent aromatic rings, which are found in the case of anthocyanins [59].

Additionally, the other transmittance peaks observed in both the CH films and the NP-containing films suggest interactions between the chitosan, biomolecules, and NPs. The results are displayed in Figure 3b–d. Figure 3b corresponds to the CH film, Figure 3c to the 1% NiZnFe_2_O_4_ NP-containing CH film, and Figure 3d to the 2% NiZnFe_2_O_4_ NP-containing CH film. The peaks corresponding to stretching and bending are listed in Table 2.

### 4.4. Color Analysis

Anthocyanins are naturally potent antioxidants, found widely in many fruits and vegetables. These compounds are responsible for providing red, purple, and blue colors to a variety of plants and have been extensively studied due to their beneficial health properties, as well as serving as coloring agents. Indeed, when incorporated into chitosan films, anthocyanins generally cause the color to shift from the yellowish to the reddish spectrum [60]. This shift in the peak is due to changes at the structural level in the anthocyanin molecules as a function of the pH and its interactions with the polymer matrix of the film [61]. In this analysis, due to their chemical nature, the anthocyanins caused a yellowish tint in the films. This yellowing tint was slightly distorted in the films after adding the NiZnFe_2_O_4_ NPs, which led to a brick-red color due to the ferrite structure. The red color of the ferrite nanoparticles interacted with the matrix of the film and resulted in very slight coloration. The shift towards red is reflected in the *a** value, as a higher *a** value means a greater red shift. For the CH films, the *a** value was 10.949 ± 0.875, while, for the 2% NiZnFe_2_O_4_ NP-containing films, it was 12.249 ± 0.976.

The color analysis resulted in the values shown in Table 3. These were obtained with the calculation of ∆E, a standard color difference indicator used to determine changes in hue and saturation. The films containing the NiZnFe_2_O_4_ NPs showed lower ∆E values than the pure chitosan films, with a smaller degree of color change [14]. In particular, the films incorporated with 1% and 2% NiZnFe_2_O_4_ NPs had ∆E values of 29.603 ± 2.332 and 29.881 ± 1.993, respectively, which were significantly lower than those of the control CH films, which were 31.797 ± 1.742. A smaller ∆E would indicate that the NPs stabilize the color profile of the films, most likely through a combination of physical and chemical interactions in the chitosan matrix, which would moderate the more severe color shift caused by the anthocyanins alone.

### 4.5. Film Thickness

The film thickness is an important parameter that significantly influences various properties of polymer films, from their mechanical strength to their optical clarity [62]. The thickness of the film is controlled mainly by the quantity of the polymeric solution used and the subsequent drying process. Consistent thickness control is highly important when mechanical integrity and optical properties are desired in films, e.g., for applications where film uniformity and precision are crucial. For example, variations in thickness will result in a difference in the transparency of light, which is vital for optical applications. Table 3 contains the measured thickness values. The thickness of the CH film was 320 ± 5 mm, while those of the films with 1% and 2% NiZnFe_2_O_4_ NPs were 330 ± 9 mm and 330 ± 6 mm, respectively. This implies that the NPs only made a small contribution to the overall thickness of the films. This consistency could indicate that the mode of incorporation of the NPs into the films does not result in a considerably altered volume or distribution in the polymer matrix when subjected to the drying process, and this is the determining factor regarding the film thickness.

### 4.6. Moisture Content, Water Solubility, and Degree of Swelling

The MC, WS, and DS are among the critical parameters that determine the applicability of a given polymer in various environmental conditions. These metrics are crucial in estimating the degradability and application suitability of polymers, particularly with respect to their environmental stability and interactions with water.

Table 4 contains the recorded %MC, %WS, and %DS values for the tested samples. For the films containing 1% NiZnFe_2_O_4_ NPs, there was a noticeable change: the moisture content decreased slightly to 6.334 ± 0.373%, while the water solubility increased significantly to 61.352 ± 2.333%, and the degree of swelling also rose to 13.526 ± 1.027%. Conversely, the 2% NiZnFe_2_O_4_ NP-containing CH films showed an interesting pattern, with moisture content of 7.619 ± 0.279%, which was higher than that of the base CH film, as well as water solubility of 54.639 ± 2.861% and a degree of swelling of 8.247 ± 0.295%. The nanoparticles’ reinforcement led to an increase in WS compared with the CH film [63]. The %WS results confirm that the nanoparticle-containing films are better suited for use in dry applications because they tend to swell more in the presence of water, which will cause them to degrade faster. Although these films have higher water solubility, they can still cause a delay in the onset of corrosion. Furthermore, the coating of materials can be achieved with such films, particularly where the easy removal of the protective coating is required before use. The studied films offer this property as harsh chemicals are not required to remove them.

### 4.7. Water Vapor Permeability

Water vapor permeability is one of the most critical parameters reflecting diffusion through a material; hence, it significantly influences coating and packaging applications where an effective moisture barrier is required. Furthermore, films with better water resistance are more effective in protecting the coated/packed material from wear and corrosion caused by water contact. In the present work, we studied the WVP of the CH films and those containing 1% and 2% NiZnFe_2_O_4_ NPs. 

In this work, it was discovered that, after 48 h, the water vapor that permeated through the CH film amounted to 4.05 ± 0.033 × 10^−10^ g⋅mm/m^2^⋅d⋅kPa. The films containing 1% NiZnFe_2_O_4_ NPs enabled 2.572 ± 0.036 × 10^−10^ g⋅mm/m^2^⋅d⋅kPa of water vapor to pass through. Then, the results declined, with the 2% NPs yielding a value of 1.732 ± 0.009 × 10^−10^ g⋅mm/m^2^⋅d⋅kPa, as shown in Table 5. This trend indicates a systematic decrease in the water vapor permeability of the films with an increase in the concentration of NPs in the film’s structure [64]. The NPs filled the voids in the polymer matrix, causing a reduction in permeability. By occupying these voids, they inhibited the passage of the water vapor molecules, hence increasing the barrier properties of the film. This mechanism is reflected in the results, where higher loadings of NPs resulted in more significant drops in the water vapor transmission rate. The *S* value (orientation or shape factor) was found to be 0.702. The aspect ratio of the NiZnFe_2_O_4_ NPs in the 1% NiZnFe_2_O_4_ NP-containing film was found to be 72.955, whereas, for the 2% NiZnFe_2_O_4_ NPs, the aspect ratio was found to be 116.61. A change in the aspect ratio can be caused by various factors, e.g., changes in the particle size or the morphology of the particles [65]. Inside the polymeric structure, various factors can impact the size and morphology of the nanoparticles—for instance, their concentration, their molecular weight, or the acid concentration [66]. Moreover, if a plant extract is used, this can impact the morphology of the nanoparticles as well [67]. In this method, we used lemon juice and lemon peel extract, which can significantly impact the nanoparticle morphology, while their magnetic nature also plays a role. Our hypothesis is that the magnetic properties of the nanoparticles and the use of lemon peel and lemon juice, which all impact the morphology and distribution of nanoparticles, gave rise to the enhanced aspect ratio. Figure 7 shows images of the films in which the difference in the nanoparticle distribution can clearly be seen, represented by arrows. These high aspect ratio values are indicative of nanoparticles with a high degree of anisotropy, which plays a significant role in creating longer and more complex paths for water molecules in navigating through the film [68]. This further increases the tortuosity, which effectively reduces the diffusion rate of water by forcing it to move around the impermeable barrier formed by the nanoparticles.

### 4.8. Mechanical Properties

The mechanical properties of films prepared by incorporating plant extracts commonly exhibit a reduction due to the presence of several biomolecules that interfere with the integrity of the polymer matrix [69]. However, this modification also confers the films with a more elastic nature, which can be desirable for specific applications. The inclusion of NiZnFe_2_O_4_ NPs further complicated the mechanical behavior of the studied films. The mechanical properties of CH films are highly dependent on the influence of the NPs distributed within them. With a homogeneous distribution, the NPs will generally increase the mechanical strength with increased material brittleness [64,70]. This is due to the uniformly dispersed NPs, which strengthen the polymer matrix itself, similarly to the effects of fibers in composite materials. In contrast, an inhomogeneous distribution of NPs tends to enhance the elasticity of films due to the localized agglomeration of the NPs, which will form regions of differential stress; these deform more easily under mechanical strain [71]. The mechanical properties play a pivotal role in the application of films. A much stronger film can be used as a storage bag, while films with lesser mechanical strength can be used as coating materials.

According to the obtained results, the control CH films, which had no NPs embedded within them, could bear stress amounting to approximately 1.52 MPa and showed plastic deformation with a maximum potential of 0.25. On the other hand, there was an increase in elasticity in the NiZnFe_2_O_4_ NP-containing films, as reflected in their behavior under mechanical testing. These findings are clearly depicted in Figure 8. The films with 2% NiZnFe_2_O_4_ NPs were mechanically stronger than those with 1% NiZnFe_2_O_4_ NPs, with stress values of 0.41 MPa and 1.04 MPa, respectively. For this reason, according to the studies by Díez-Pascual et al. [64] and Jeon, I.-Y et al. [71], the 2% NiZnFe_2_O_4_ NPs films may have possessed a more homogeneous distribution compared to the 1% NiZnFe_2_O_4_ NPs films, because the latter had poorer mechanical properties, as indicated in Figure 8. Table 4 contains the E and %E values recorded for the samples.

As shown in Table 6, a brief literature review was conducted to compare our developed films and their properties with those of other researchers, including aspects such as the color analysis, WVP, mechanical properties, %WS, %MC, and %DS. It was found that our developed CH films with LPE showed a TS value of 1.52 MPa, with 38 times higher tensile strength compared to the CH film developed by Jancikova, which contained *Clitoria ternatea* and had a TS value of 0.04 MPa [72].

### 4.9. Puncture Strength

The puncture strength is of great importance regarding the mechanical properties of films, reflecting their surface strength and resistance to penetrating forces. It is essential is determining the practical resilience and durability of film materials under concentrated load conditions, serving to indicate whether a film can handle higher friction. For the plain CH films, the measured puncture strength was around 0.24 MPa. However, upon the addition of the NiZnFe_2_O_4_ NPs, this property was improved significantly. The films with 1% NiZnFe_2_O_4_ NPs exhibited puncture strength of almost 0.28 MPa, while the value for the films containing 2% NPs was much higher, at approximately 0.39 MPa. The results obtained are plotted in Figure 9. The higher puncture strength of the NP-modified films suggests improved surface strength and a more elastic nature before rupture [70]. The application of localized stress in puncture testing indicates whether films could withstand sharp or pointed stresses and friction in real-world use.

Therefore, these findings indicate that the NiZnFe_2_O_4_ NPs effectively reinforced the CH film matrix, thereby distributing and absorbing the applied forces more efficiently. This resulted in a film that could sustain greater force without failure and showed improved pointwise elasticity under specific stress conditions compared to the baseline CH films.

## 5. Conclusions

This study aimed to develop CH-based green polymeric films by utilizing 100% naturally sourced juices and extracts. Lemon juice served as the dissolution medium for chitosan, acting as a replacement for analytical-grade citric acid, while lemon peel extract was used both to synthesize the NiZnFe_2_O_4_ NPs and to enhance the CH film’s properties. The XRD analysis confirmed the crystalline phase of the NPs, indicating successful synthesis, with an average crystallite size of ~29 nm. The NPs were incorporated into the polymeric matrix to examine their impacts on the films’ mechanical properties. The incorporation of 1% NiZnFe_2_O_4_ NPs resulted in a decrease in mechanical strength but a significant increase in elasticity, being approximately 90% greater than that of the plain CH films. Conversely, the films containing 2% NiZnFe_2_O_4_ NPs demonstrated enhanced mechanical strength, attributed to the more homogeneous distribution of the NPs within the matrix, but their tensile strength decreased. Due to the presence of LPE, a yellowish hue was observed in the films, and the *a** value of the bare CH films was 10.949 ± 0.875, while higher concentrations of NPs introduced a stronger reddish shade due to their naturally darker color. Furthermore, the WVP data showed that the NPs resulted in improved barrier properties, making them suitable candidates for coating materials to delay wear and corrosion, as well as providing protection from water and air in controlled environments. Meanwhile, the WS values show that the films can be degraded more easily, making them a sustainable and eco-friendly alternative to the widely used hazardous plastics. Future research should test these films in real-world tribological scenarios to validate their practical applications.

### Project Details/Statement

This study is part of a Ph.D. research project initiated for the development of chitosan films from naturally extracted juices, aimed at studying their wide range of applications and improving their mechanical properties through nanoparticle reinforcement, including nickel iron oxide, zinc iron oxide, nickel–zinc iron oxide, and other metal oxides. This will be published when the work is complete, reporting their impacts on the mechanical properties of polymeric films. This work is anticipated to contribute to sustainable development and global efforts to fight climate change, and it is a continuation of the work “Environmentally Sustainable and Ecofriendly Method for Chitosan Films Synthesis Using Natural Acids and Impact of Zinc Ferrite Nanoparticles on Water Solubility and Physical Properties”. This project’s uniqueness lies in its highly innovative and environmentally benign method of material synthesis, using naturally extracted lemon juice and residual lemon peel extract.

## Figures and Tables

**Figure 1 polymers-16-03455-f001:**
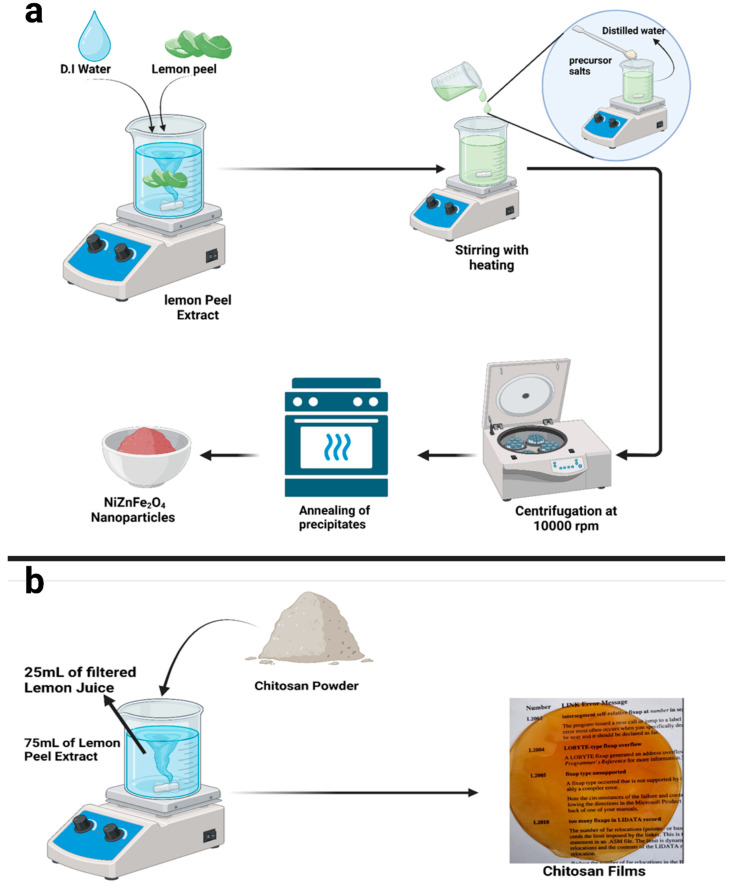
Schematic of (**a**) NiZnFe_2_O_4_ nanoparticle synthesis using lemon peel extract and (**b**) development of NP-containing CH films.

**Figure 2 polymers-16-03455-f002:**
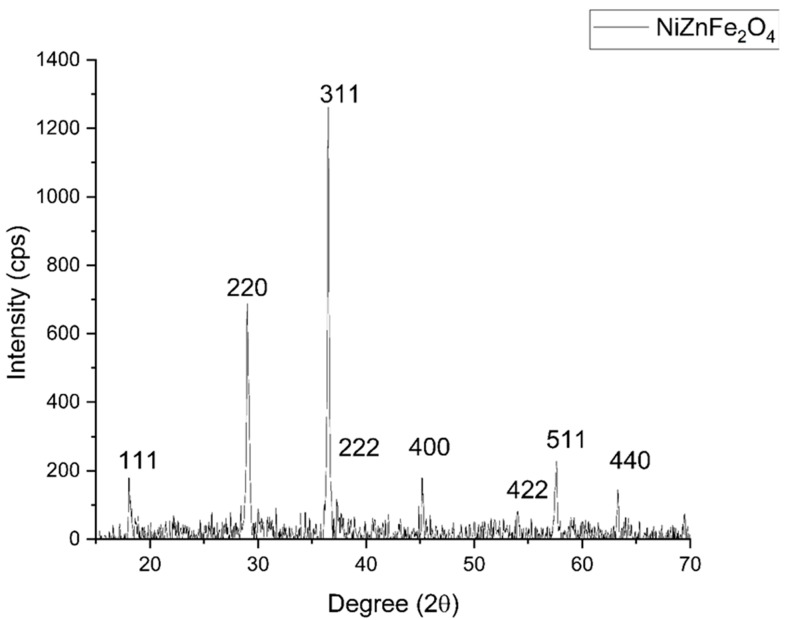
XRD pattern obtained for NiZnFe_2_O_4_ NPs.

**Figure 3 polymers-16-03455-f003:**
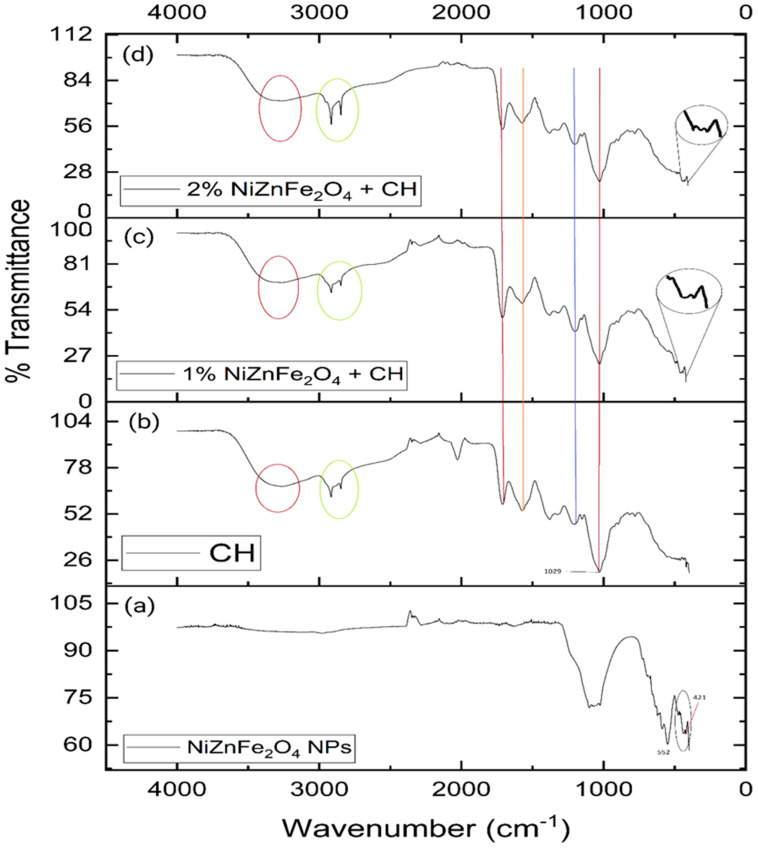
FTIR results: (**a**) NiZnFe_2_O_4_ NPs, (**b**) CH film, (**c**) 1% NiZnFe_2_O_4_ NP-containing CH film, and (**d**) 2% NiZnFe_2_O_4_ NP-containing CH film.

**Figure 4 polymers-16-03455-f004:**
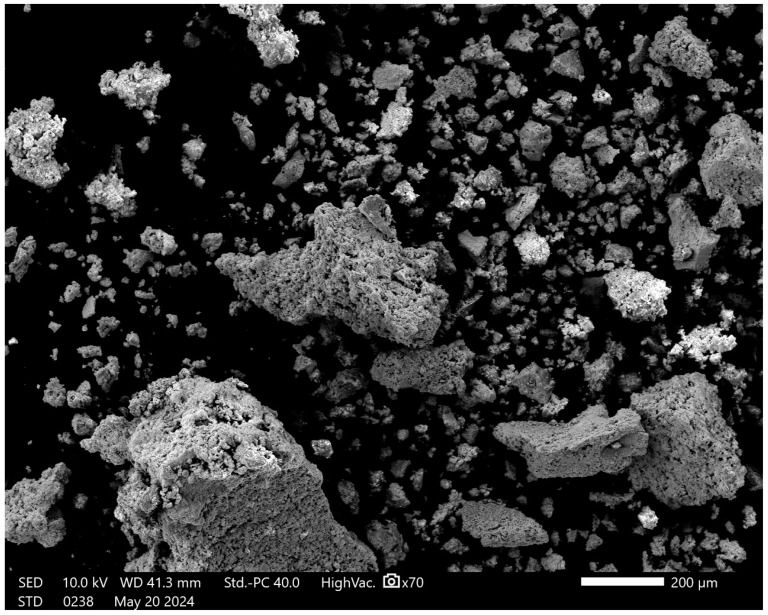
High-resolution SEM image of NiZnFe_2_O_4_ NPs.

**Figure 5 polymers-16-03455-f005:**
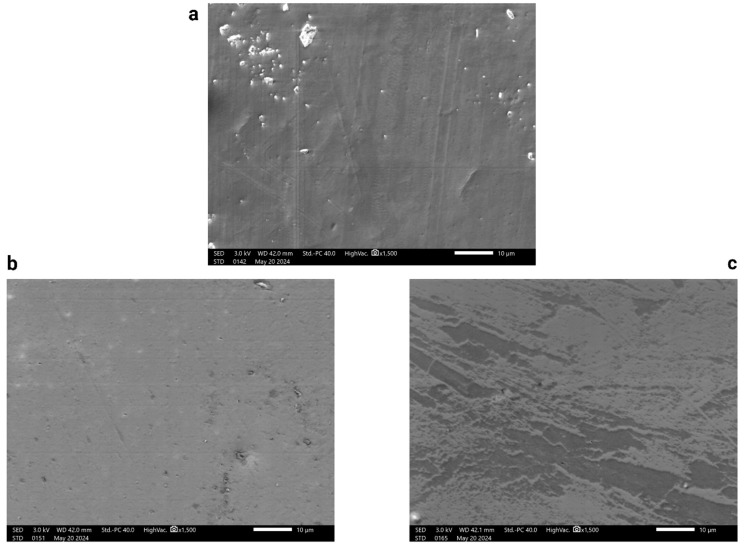
SEM images: (**a**) a captured image of the CH film; (**b**,**c**) images of the 1% and 2% NiZnFe_2_O_4_ NP-containing CH films.

**Figure 6 polymers-16-03455-f006:**
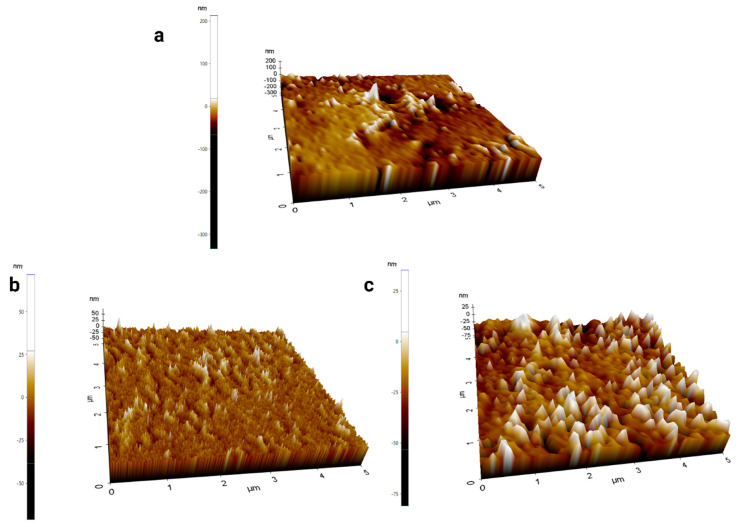
AFM images of CH films: (**a**) CH film, (**b**) 1% NiZnFe_2_O_4_-containing CH film, and (**c**) 2% NiZnFe_2_O_4_-containing CH film.

**Figure 7 polymers-16-03455-f007:**
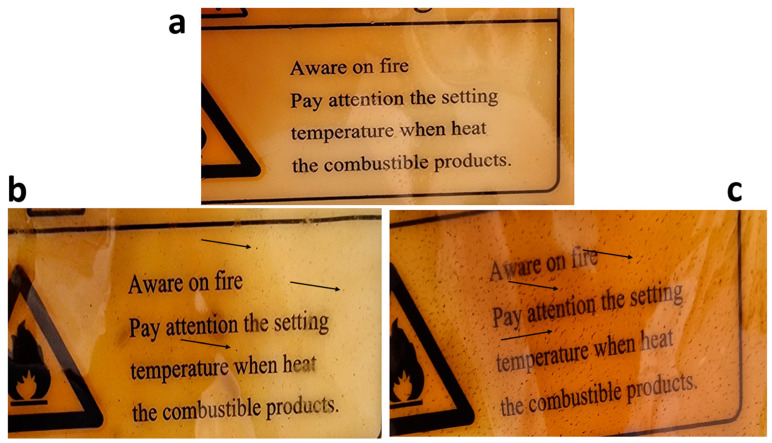
Magnified views of the (**a**) CH and (**b**,**c**) 1% and 2% NP-containing CH films, showing the impact of the nanoparticles on the films.

**Figure 8 polymers-16-03455-f008:**
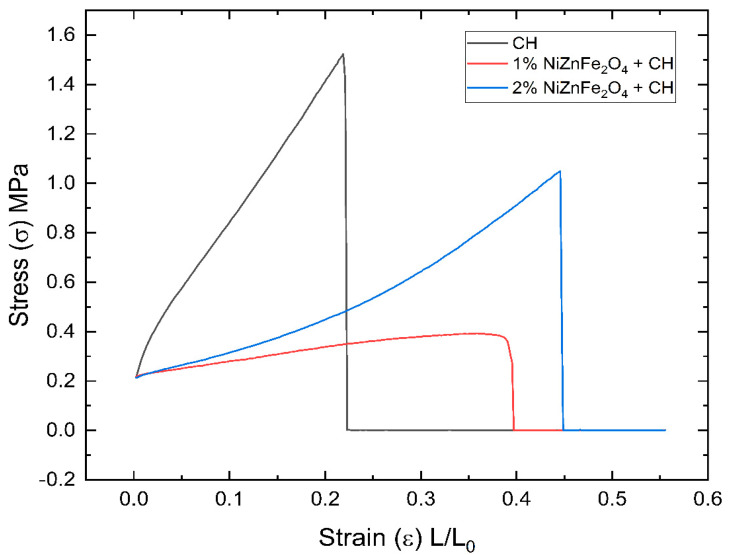
Mechanical property results obtained for CH and NiZnFe_2_O_4_ NP-containing CH films.

**Figure 9 polymers-16-03455-f009:**
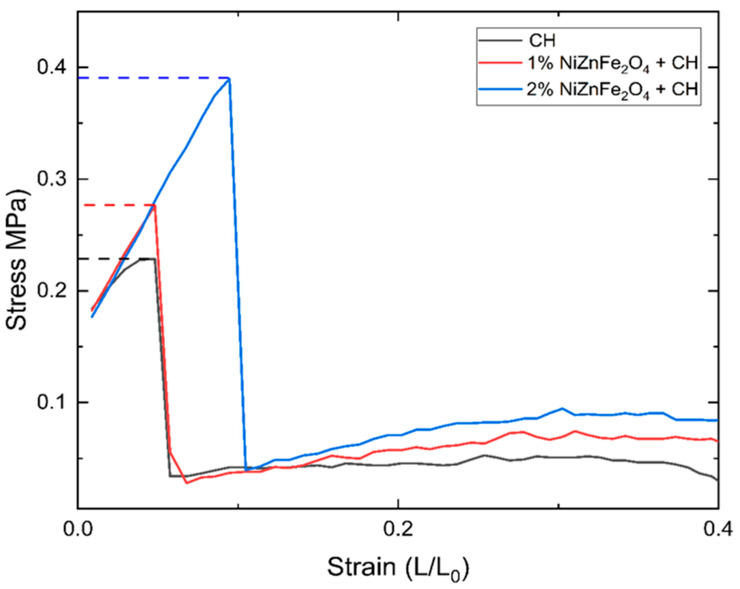
Puncture strength results obtained for CH and NiZnFe_2_O_4_ NP-containing CH films.

**Table 1 polymers-16-03455-t001:** Crystallite size calculated for NiZnFe_2_O_4_ NPs using Scherrer’s formula.

hkl	Peak Position	FWHM	Crystallite Size (nm)
111	18.112	0.240	34.98
220	29.046	0.313	27.36
311	36.502	0.325	26.89
222	36.905	0.213	41.04
400	45.940	0.397	22.68
422	54.891	0.350	26.69
511	57.586	0.358	26.43
440	63.310	0.402	24.23
Average			28.787

**Table 2 polymers-16-03455-t002:** FTIR peaks and corresponding bending, stretching, and vibration types.

Peak (cm^−1^)	Correspondence	Peak (cm^−1^)	Correspondence
421—NiZnFe_2_O_4_ NPs	M–O vibration	1629—NiZnFe_2_O_4_ NPs	O–H bending
522—NiZnFe_2_O_4_ NPs	M–O vibration	1708—CH + NPs Film	C=O stretching
785—CH + NPs Film	C–H bending	2110—NiZnFe_2_O_4_ NPs	C≡C stretching
896—CH + NPs Film	C=C bending	2850—CH + NPs Film	N–H stretching
1029—CH + NPs Film	S=O stretching	2914—CH + NPs Film	N–H stretching
1101—NiZnFe_2_O_4_ NPs	C–H bending	2985—NiZnFe_2_O_4_ NPs	C–H stretching
1220—CH + NPs Film	C–O stretching	3280—NiZnFe_2_O_4_ NPs	O–H stretching
1550—CH + NPs Film	N–O stretching	3289—CH + NPs Film	O–H stretching

**Table 3 polymers-16-03455-t003:** Color analysis results for CH and NiZnFe_2_O_4_ NP-containing CH films, where *p* < 0.05.

Sample	*L**	*a**	*b**	∆E	Thickness (µm)
**CH**	49.766 ± 2.517	10.949 ± 0.875	29.550 ± 1.068	31.797 ± 1.742	320 ± 5
**1% NPs**	47.916 ± 1.693	11.841 ± 1.009	26.621 ± 2.398	29.603 ± 2.332	330 ± 9
**2% NPs**	51.721 ± 3.037	12.249 ± 0.976	28.285 ± 1.055	29.881 ± 1.993	330 ± 6

**Table 4 polymers-16-03455-t004:** The %MC, %WS, and %DS values (where *p* < 0.05) and the E (MPa) and %E values observed for the CH and NiZnFe_2_O_4_ NP-containing CH films.

Sample	% MC	% WS	% DS	E (MPa)	%E
**CH**	7.228 ± 0.460	37.662 ± 1.867	9.740 ± 0.433	6.4	15.183
**1% NiZnFe_2_O_4_**	6.334 ± 0.373	61.352 ± 2.333	13.526 ± 1.027	1.026	31.264
**2% NiZnFe_2_O_4_**	7.619 ± 0.279	54.639 ± 2.861	8.247 ± 0.295	2.311	36.833

**Table 5 polymers-16-03455-t005:** WVP values (with *p* < 0.05) observed for the CH and NiZnFe_2_O_4_ NP-containing CH films after 72 h.

Sample Name	Weight of Water Permeating Through the Film (×10^−10^ g⋅mm/m^2^⋅d⋅kPa)		
	24 h	48 h	72 h
**CH**	1.875 ± 0.027	2.837 ± 0.013	4.05 ± 0.033
**1% NiZnFe_2_O_4_**	1.193 ± 0.013	2.248 ± 0.009	3.012 ± 0.036
**2% NiZNFe_2_O_4_**	0.737 ± 0.014	1.692 ± 0.008	2.732 ± 0.009

**Table 6 polymers-16-03455-t006:** Literature review.

Polymer	Reinforcement	*L**	*a**	*b**	%WS	%DS	%MC	WVP (g⋅mm/m^2^⋅d⋅kPa)	TS (MPa)	%E	Ref.
CH	*Clitoria* *ternatea*	-	-	-	-	-	-	-	0.04	108	[72]
CH	Lecithin + tea tree essential oil	-	-	-	19.26	-	-	4.3 × 10^−11^	1.54	317.33	[73]
CH	Rosemary	66	−4.9	14	28	170	24	-	12	44	[17]
CH	Glycerol 30%	-	-	-	-	-	-	-	6	32	[23]
CH	LPE + 2% NiZnFe_2_O_4_	51.721	12.249	28.285	7.619	54.639	8.247	1.732 × 10^−10^	1.04	36.833	Current Work

## Data Availability

The original contributions presented in the study are included in the article, further inquiries can be directed to the corresponding author.

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
