# Peer review of "The Impact of Nickel–Zinc Ferrite Nanoparticles on the Mechanical and Barrier Properties of Green-Synthesized Chitosan Films Produced Using Natural Juices"

_polymers, 2024, doi:10.3390/polym16243455_

Round 1
Reviewer 1 Report
Comments and Suggestions for Authors
Impact of Nickel Zinc Ferrite Nanoparticles on the Mechanical and Barrier Properties of Green Synthesized Chitosan Films using Natural Juices
1. A minor English revision is required.
2. The introduction seems to be too short; it is recommended to add extensive review of the latest similar works and their findings. Adding few case studies would be beneficial addition for the overall impact of the work
3. Novelty of the paper is not clear to me; I would suggest to give a more emphasis on the defining the novelty.
4. For the section 2.6.2, it is mentioned that the cups were sealed, please define, how were the cups sealed? Was any adhesive utilized for the sealing purpose? Was any blank test conducted so as to make sure there is no leakage?
5. How was the relative humidity of 75% maintained in a desiccator and what was the temperature?
6. Also measuring a WVP for three days is not a justifiable result. The test must at least be carried out for 10 days.
7. Figure 4 reveals the shape of the particles; it is evident that these particles have the size of more than 200 microns even some go beyond 1000 microns. It is not clear to me, why do authors refer these particles as ‘Nanoparticles’? This size could have easily been seen in the simple microscope.
Furthermore, these even don’t have a uniform shape, how are they supposed to block diffusing water molecules and give strength to films?
8. To block water molecules from diffusing, the particle must have a high aspect ratio; therefore, author must calculate the aspect ratio of these particles and then do the needful justifications via applying Bhardwaj’s permeability model.
Please refer to below literature for calculating Bharadwaj’s permeability model: https://doi.org/10.3390/membranes12070701.
Comments on the Quality of English Language
Needs minor revision.
Author Response
We sincerely thank the reviewers for their invaluable time and effort in reviewing our manuscript. We appreciate the thoughtful insights and constructive feedback they provided, especially considering their busy schedules. Their input has significantly enhanced the readability and impact of our work, helping us present a more compelling and well-structured study. These suggestions and comments have not only helped us with this manuscript but will help us for future research endeavors also. We greatly appreciate the thorough and prompt review, especially given our approaching graduation deadline in the third week of November. Your feedback has allowed us to strengthen our study, and we are grateful for your efforts to expedite this process.
The following are your kind suggestion and constructive comments and our reply to them.
Reviewer 1
Comment: minor English revision is required.
Reply: Esteemed reviewer, Thank you for your comment and confidence. We have made sure of grammatical changes in the manuscript.
Comment: The introduction seems to be too short; it is recommended to add extensive review of the latest similar works and their findings. Adding few case studies would be beneficial addition for the overall impact of the work.
Reply: Dear Reviewer, we have added to the introduction, making it more knowledgeful by adding various case studies. Thank you for letting us polish our manuscript more.
Comment: Novelty of the paper is not clear to me; I would suggest to give a more emphasis on the defining the novelty.
Reply: Dear Reviewer, the method is novel because it uses lemon juice as a replacement for dissolution of chitosan, as there is no report on it before. Further novelty is its sustainability, as we are using the left-over peels to obtain lemon peel extract for synthesis of NiZnFeâ‚‚Oâ‚„ nanoparticles. we have added
“This greener approach makes this method novel and unique, as there has been no studies that used LJ as replacement to acetic acid for the dissolution of CH. Furthermore, this method is also sustainable with being novel, as we are utilizing lemon peels to obtain LPE, for the synthesis of reinforcing material i.e., NiZnFeâ‚‚Oâ‚„ nanoparticles”
Comment: For the section 2.6.2, it is mentioned that the cups were sealed, please define, how were the cups sealed? Was any adhesive utilized for the sealing purpose? Was any blank test conducted so as to make sure there is no leakage?
Reply: Dear reviewer, thank you for your critical comment, for sealing we used vacuum grease, whereas the cups were sealed by tightening a steel ring over the cups with clips to get better sealing. We have added
“… and film samples were sealed over the cup surfaces to ensure leak-proof conditions, using vacuum grease, further covered with steel ring tightened with the help of screws to provide better sealing”
Comment: How was the relative humidity of 75% maintained in a desiccator and what was the temperature?
Reply: Esteemed reviewer, the desiccator was placed inside a controlled environmental chamber. We have added
“The film-sealed cups were then placed in a desiccator with RH = 75%, by placing the desiccator in a controlled environment chamber (Thermo Scientific, 3911 – forma environmental chamber).”
Thank you for helping us with making the procedures clearer.
Comment: Also measuring a WVP for three days is not a justifiable result. The test must at least be carried out for 10 days.
Reply: Dear reviewer, Thank you for your comment cum suggestion. Due to lack of time and resources, we could not perform this study at this particular time. Furthermore, various studies did this experiment for 3 days, to study the trend, which was our objective also (citation 46), while for future studies, we will definitely take guidance from your valuable suggestion.
Comment: Figure 4 reveals the shape of the particles; it is evident that these particles have the size of more than 200 microns even some go beyond 1000 microns. It is not clear to me, why do authors refer these particles as ‘Nanoparticles’? This size could have easily been seen in the simple microscope. Furthermore, these even don’t have a uniform shape, how are they supposed to block diffusing water molecules and give strength to films?
Reply: Esteemed reviewer, The SEM shows agglomerated nanoparticles, not the nanoparticles themselves. Due to uncoated surface of nanoparticles, when we tried to decrease the lens to sample distance, the nanoparticles started moving because of electron gun’s charge. Therefore, the pictures were captured of the clusters rather than nanoparticles. To make the results better readable, we have added
“NPs predominantly displayed a clustered arrangement. This clustering is primarily attributed to the intrinsic magnetic properties of the NPs, which promote aggregation. Furthermore, the NPs exhibited distinct self-assembling behavior, organizing themselves into porous structures. These pores are not merely surface features but are integrated throughout the clusters, suggesting a complex internal structure.”
We thank you for your help in making this manuscript better readable.
Comment: To block water molecules from diffusing, the particle must have a high aspect ratio; therefore, author must calculate the aspect ratio of these particles and then do the needful justifications via applying Bhardwaj’s permeability model.
Please refer to below literature for calculating Bharadwaj’s permeability model: https://doi.org/10.3390/membranes12070701
Reply: Dear reviewer, thank you for bringing it to our attention. The suggested manuscript and another manuscript with title “Process Parameter Optimization of a Polymer Derived Ceramic Coatings for Producing Ultra-High Gas Barrier” were of great help. We have added the data. Thank you for helping with bringing more to the table for the readers.
Reviewer 2 Report
Comments and Suggestions for Authors
This study explores the synthesis of chitosan (CH) films enhanced with lemon juice (LJ) and reinforced with NiZnFeâ‚‚Oâ‚„ nanoparticles (NPs) to create a sustainable, corrosion-resistant coating for medical instruments. By incorporating NiZnFeâ‚‚Oâ‚„ NPs, the researchers improved the mechanical strength and barrier properties of the biopolymer films. These CH films demonstrated effective corrosion protection, offering an environmentally friendly alternative to traditional coatings for medical applications, aligning with sustainability goals by utilizing renewable materials.
Comments:
1. Please add the results of statistical analysis (p-values) in Table 3, 4 and 5.
2. Figure 1 is confusing. When were the nanoparticles added into chitosan films? I suggest using two figures (Figure 1a and 1b) to show two procedures of fabrication (NP and Chitosan-NP film) separately.
3. The Figure 4 shows many large clusters of NiZnFe2O4. Could author have any result of size distribution of particles.
4. Section 4.2, is there any quantification analysis for surface roughness?
5. Section 2.6.4, is this method developed by authors or from other studies?
6. Section 2.3, please add reference for equation 1.
7. Section 2.1, were all lemons bought and processed at the same time? If not, how did authors maintain the quality of juice/extrication? Do they combine all juice/extrication together for next step?
8. Is NiZnFe2O4 nanoparticles approved by FDA? Also, please provide more references about its application.
9. I suggest using the full name for lemon juice (not LJ).
Author Response
We sincerely thank the reviewers for their invaluable time and effort in reviewing our manuscript. We appreciate the thoughtful insights and constructive feedback they provided, especially considering their busy schedules. Their input has significantly enhanced the readability and impact of our work, helping us present a more compelling and well-structured study. These suggestions and comments have not only helped us with this manuscript but will help us for future research endeavors also. We greatly appreciate the thorough and prompt review, especially given our approaching graduation deadline in the third week of November. Your feedback has allowed us to strengthen our study, and we are grateful for your efforts to expedite this process.
The following are your kind suggestion and constructive comments and our reply to them.
Comment: Please add the results of statistical analysis (p-values) in Table 3, 4 and 5.
Reply: Dear Reviewer, thank you for your kind comment, and for bringing it to our attention. We have added the p-values to respective tables.
Comment: Figure 1 is confusing. When were the nanoparticles added into chitosan films? I suggest using two figures (Figure 1a and 1b) to show two procedures of fabrication (NP and Chitosan-NP film) separately
Reply: Dear Reviewer, thank you for your suggestion. We have divided the Figure 1 in Figure 1a and 1b.
Comment: The Figure 4 shows many large clusters of NiZnFe2O4. Could author have any result of size distribution of particles
Reply: Esteemed Reviewer, Thank you so much for your critical comment. We could not do Size analysis due to the non-availability of zeta size and other instruments. Currently, due to shortage of time, and lack of access to the lab and sample, we cannot perform this study. But your valuable comments will be kept in sight for future studies.
Comment: Section 4.2, is there any quantification analysis for surface roughness?
Reply: Dear Reviewer, Thank you again for this valuable and insightful comment. We have analyzed the surface roughness, using SEM and AFM surface analysis, but have not performed any (RMS) study. Due to no access to the lab, we will keep this suggestion in mind and will surely enhance future research.
Comment: Section 2.6.4, is this method developed by authors or from other studies?
Reply: Dear Reviewer, Thank you so much for such detailed review. Thank you for your comment. Yes, we previously employed method and have cited it by mentioning a sentence “A pre-employed method was followed for this study”
Comment: Section 2.3, please add reference for equation 1.
Reply: Dear Reviewer, Thank you for your critical review of our manuscript. A reference has been provided.
Comment: Section 2.1, were all lemons bought and processed at the same time? If not, how did authors maintain the quality of juice/extrication? Do they combine all juice/extrication together for next step?
Reply: Esteemed Reviewer, all the lemons were processed for juice extraction at the same time, and the juice was used within 2 weeks. While maintaining the quality of juice, it was stored in an airtight storage bottle at 4 °C at all times.
Comment: Is NiZnFe2O4 nanoparticles approved by FDA? Also, please provide more references about its application
Reply: Dear Reviewer, NiZnFe2O4 itself is not FDA approved, but is being studied for potential application in many fields, and according to your suggestion we have added those in the introduction section
“NiZnFe2O4 NPs is also known for its magnetic properties and biomedical applications [30]. They have been used for their potential super capacitance [31], electronic application [32], drug loading for cancer therapy [33], and antimicrobial applications [34].”
Comment: I suggest using the full name for lemon juice (not LJ).
Reply: Esteemed Reviewer, thank you for your suggestion and we have incorporated the suggestion throughout the manuscript for lemon juice.
Round 2
Reviewer 1 Report
Comments and Suggestions for Authors
Why do the same particles change their aspect ratio by changing their concentration in the film?
Measuring the particles' SEM imaging is recommended to justify/verify the calculated aspect ratio i.e. ~72 and 116. In addition, a cross-section of the film is also recommended for verifying the orientation of the dispersed particles..
Comments on the Quality of English LanguageA minor English check is necessary.
Author Response
Dear Review, thank you once again for the timely review of the manuscript. Your previous comments helped us greatly to improve the manuscript quality and readability. And your current comment has also been a brainstorming session and has helped us to improve the manuscript quality more effectively. We can understand that it takes time to review a manuscript, and we are thankful to you for investing your time and energy in helping us improve the manuscript quality.
Here are our responses to your critical comments:
Comment: Why do the same particles change their aspect ratio by changing their concentration in the film?
Measuring the particles' SEM imaging is recommended to justify/verify the calculated aspect ratio i.e. ~72 and 116. In addition, a cross-section of the film is also recommended for verifying the orientation of the dispersed particles.
Response: Dear Reviewer, thank you for your speedy review of our manuscript. We are thankful, as time is if essence for us. Since this manuscript is a PhD requirement and the student is to graduate in a week’s time. To answer your comment on why same nanoparticles change their aspect ratio, in varying concentrations. Our hypothesis is the magnetic nature. But there are other factors that could impact the nanoparticle morphology also. For example the nanoparticles tend to increase in size when put in an acidic medium, and since we were using lemon juice it could impact the nanoparticle morphology. Due to higher concentration, the nanoparticle agglomerate better hence the aspect ration changes. Therefore we have added to the manuscript:
“The change in aspect can be caused by various factors, i.e., change in particle size, and morphology of the particle [65]. Inside the polymeric structure various factors can impact on the size and morphology of the nanoparticles, for instance their concentration, molecular weight, and acid concentration [66]. And if a plant extract is used, it can impact the morphology of the nanoparticles also [67]. Since in this method, we have used lemon juice and lemon peel extract, they can largely impact the nanoparticle morphology, while their magnetic nature plays its role also. Our hypothesis is that due to magnetic properties of the nanoparticles and the use of lemon peels and lemon juice is impacting the morphology and distribution of the nanoparticles, hence giving rise to enhanced aspect ratio. Figure 7 shows film photos in which the difference in nanoparticle distribution can easily be seen.”

Coming to your recommendation for SEM. We greatly appreciate your recommendation, and we agree with its importance, but we do not have access to SEM at this point in time, but we will surely use this recommendation in our upcoming projects. And we are sure that this information will largely help us in future endeavors. We hope our answers will be satisfactory.
Round 3
Reviewer 1 Report
Comments and Suggestions for Authors
Significant improvement is being made. Also, the explanation is fairly acceptable.